# How often do leading biomedical journals use statistical experts to evaluate statistical methods? The results of a survey

Tom E. Hardwicke[1], Steven N. Goodman[2,3,4]*

1 Meta-Research Innovation Center Berlin (METRIC-B), QUEST Center for Transforming Biomedical Research, Berlin Institute of Health, Charité –Universitätsmedizin Berlin, Berlin, Germany, 2 Meta-Research Innovation Center at Stanford (METRICS), Stanford, CA, United States of America, 3 Department of Epidemiology & Population Health and Medicine, Stanford University, Stanford, CA, United States of America, 4 Department of Medicine, Stanford University, Stanford, CA, United States of America

* steve.goodman@stanford.edu

**Data Availability Statement:** All relevant data are available within: https://osf.io/a43ut/.

**Funding:** The author(s) received no specific funding for this work.

## Abstract

Scientific claims in biomedical research are typically derived from statistical analyses. However, misuse or misunderstanding of statistical procedures and results permeate the biomedical literature, affecting the validity of those claims. One approach journals have taken to address this issue is to enlist expert statistical reviewers. How many journals do this, how statistical review is incorporated, and how its value is perceived by editors is of interest. Here we report an expanded version of a survey conducted more than 20 years ago by Goodman and colleagues (1998) with the intention of characterizing contemporary statistical review policies at leading biomedical journals. We received eligible responses from 107 of 364 (28%) journals surveyed, across 57 fields, mostly from editors in chief. 34% (36/107) rarely or never use specialized statistical review, 34% (36/107) used it for 10–50% of their articles and 23% used it for all articles. These numbers have changed little since 1998 in spite of dramatically increased concern about research validity. The vast majority of editors regarded statistical review as having substantial incremental value beyond regular peer review and expressed comparatively little concern about the potential increase in reviewing time, cost, and difficulty identifying suitable statistical reviewers. Improved statistical education of researchers and different ways of employing statistical expertise are needed. Several proposals are discussed.

## Introduction

Scientific claims in the biomedical literature are usually based on statistical analyses of data [1, 2]. However, misunderstanding and misuse of statistical methods is prevalent and can threaten the validity of biomedical research [2–8]. Statistical practices used in published research, particularly in leading journals, powerfully influence the statistical methods used by both the prospective contributors to those journals and the larger scientific community. These practices are in turn shaped by the peer review and editing process, but most biomedical peer reviewers

**Competing interests:** The authors have declared that no competing interests exist.

and editors do not have expert statistical or methodologic training. Many biomedical research journals therefore enlist statistical experts to supplement regular peer review [9], input that empirical studies have consistently shown to improve manuscript quality [10–17].

Some biomedical journals have adopted statistical review since at least the 1970s. Leading journals such as the Lancet [12], the BMJ [18], Annals of Internal Medicine [19], and JAMA [20] all employ statistical review. Two surveys, one in 1985 and another in 1998, sought to systematically characterise biomedical journal policies and practices regarding statistical review [21, 22]. Since the last survey in 1998, concerns about the validity of research findings have risen dramatically, with poor statistical practice being recognized as an important contributor [7]. We were interested to see to what extent these concerns had spurred changes among leading biomedical journals in the use of or attitudes towards statistical review.

## Methods

### Sample

From the complete list of Web of Science subject categories (228) we identified all 68 subdomains representing biomedicine. We selected the top 5 journals by impact factor within each sub-domain. We supplemented this list with 68 additional journals previously included in the survey by Goodman and colleagues [22], and assigned each of these to their relevant sub-domain. Finally, we removed any duplicates that appeared in multiple sub-domains. This resulted in a sample of 364 journals.

### Methods

The digital survey instrument (see https://osf.io/dg9ws/) was an adapted and expanded version of the survey previously conducted by Goodman and colleagues [22]. There were 16 questions in total, however the exact number presented to a respondent depended on their response to the first question: "Of original research articles with a quantitative component published in your journal, approximately what percentage has been statistically reviewed?" If respondents indicated that fraction was less than or equal to 10%, they skipped to a question about why they rarely use statistical review (Q12). If the fraction was greater than 10%, they completed a detailed series of questions relating to statistical review policies at their journal (Q2—Q11). A question about ability and willingness to use statistical review (Q13) was asked of all respondents except for those who indicated that their journal's articles rarely or never require statistical review, or that statistical aspects of the article are adequately handled during regular peer-review and/or by editors (for Q12). Three questions concerned journal characteristics (Q14—Q16). Finally, all participants were asked to share additional comments in a free-text response (Q17). The full survey instrument is available online (https://osf.io/dg9ws/). The questions and response options reported here are paraphrased for brevity. This survey was approved by Stanford IRB #42023.

### Survey procedure

The survey was developed and hosted on the 'Qualtrics' platform and distributed via e-mail. The invitation e-mail (see https://osf.io/9px8r/) outlined the purpose of the survey and included a link to the survey instrument. Depending on availability, we e-mailed either the Editor-in-Chief, the Managing Editor, or used the general journal contact address (in that order of priority). The first wave of e-mails was sent on August 9th 2017 and data collection was finalized on December 9th 2017. All respondents were told that the journal name would

not be reported or associated with any answers. Non-respondents, were sent up to three reminder e-mails as required, dispatched at approximately two-week intervals.

## Results

### Sample characteristics

We received responses from 127 (35%) of the 364 biomedical journals surveyed. Of the 68 subject areas, at least one journal responded in 57 areas, with a range from 1 to 6 journals (Median = 2, see S1 Table). Twenty respondents were excluded for providing minimal information: 11 opened the survey but did not fill it out and 9 only completed question 1. This left 107 responses (29%) suitable for further analysis. Journals were classified into types by S.N.G. based on journal contents. There were 5 basic research journals, 86 clinical research journals, 2 hybrid (basic and clinical) journals, 3 methods journals, 2 policy journals, and 9 review journals.

The vast majority of respondents identified themselves as having editorial roles: Editors-in-Chief ($n = 77/107$, 72%), managing editors ($n = 12$, 11%), deputy editors ($n = 4$, 4%), associate editors ($n = 7$, 7%), three statistics or methodology editors, one production editor, one peer review coordinator, and 2 missing descriptors.

The median number of original research articles published annually by these journals was 164 (10th-90th percentiles 48 to 300; 15 missing). Median journal acceptance rate was 18% (10th-90th percentiles 6 to 45, 6 missing).

### How frequent is statistical review?

36 (34%) of 107 respondents indicated that statistical review was used for 10% or fewer of articles, 36 (34%) for between 10% and 50% of articles, 10 (9%) for between 50% and 99% of articles, and for 25 (23%) statistical review was used for all articles (S1 Fig). Clinical and hybrid journals (N = 88) used statistical review for a greater proportion of articles (median = 30%) compared to other journal types (N = 19, median = 2%).

For the 36 journals where statistical review was rare, 14 respondents indicated that statistical review is not required for the types of articles they handle, 9 respondents said they lacked necessary resources or access to statistical reviewers, and 8 respondents indicated that statistical aspects of manuscripts are already adequately handled by regular peer review and/or by the editors. Five responses were "other" or missing.

### Ability/willingness to use statistical review

All 107 respondents were asked to rate the extent to which various factors influenced their ability/willingness to use statistical review (see Fig 1).

### Statistical review policies

Further questions about statistical review policies and procedures were only asked of the 71 journals reporting that they reviewed more than 10% of submitted articles.

**Source and training.** 56% of 71 respondents indicated that statistical reviewers are selected from members of the editorial team (Fig 2 Panel A). The median number of statistical reviewers on the editorial team was 2 (10th-90th percentile 1 to 5, 6 missing). 34% relied on a pool of external reviewers, median size 11 (10th-90th percentile, 4 to 48, 2 missing). It was uncommon to identify statistical reviewers on an ad-hoc basis (7%).

## Factors affecting ability/willingness to use statistical review

**Fig 1. Factors affecting editor's ability or willingness to use statistical review.** Responses to the question 'To what extent do the following factors affect your ability or willingness to use statistical review (or use it more) at this journal?' N = 107. Percentages sum to about 80% because 21 (20%) responses were missing.

86% of respondents indicated that most or all of their statistical reviewers have doctoral level training in a quantitative discipline (Fig 2 Panel B), and a narrow majority (55%) paid statistical reviewers (Fig 2 Panel C).

*Review logistics.* 59% of the 71 journals did not require statistical reviewers to complete a software template or ask them to follow general guidelines (Fig 3, Panel A). 31% provided guidelines, 4% provided a software template, and 4% provided both. 72% of journals using statistical reviewers "Always" or "Usually" have them see a revised version of the manuscript (Fig 3 Panel B). It was rare for statistical reviewers to never see revised manuscripts.

35% of journals solicited statistical review contemporaneously with peer review, 27% after regular peer review but before an editorial decision, 17% "ad-hoc" and 6% only after an editorial decision had been made (Fig 3 Panel C).

**Outcomes of statistical review.** The majority of respondents (73%) indicated that statistical review results in important changes to the reviewed manuscript 50% or more of the time (Fig 4 Panel A). Roughly one quarter reported a delay in decision time of zero, less than a week, 1–2 weeks and greater than 2 weeks respectively (Fig 4 Panel B).

**Value of statistical review.** Substantial majorities of respondents believed statistical review to have considerable incremental value beyond regular peer review. This extended to critical manuscript elements supporting proper conclusions, beyond statistics per se, including results interpretation, presentation, consistency of conclusions with the evidence, and the reporting of study limitations (Fig 5).

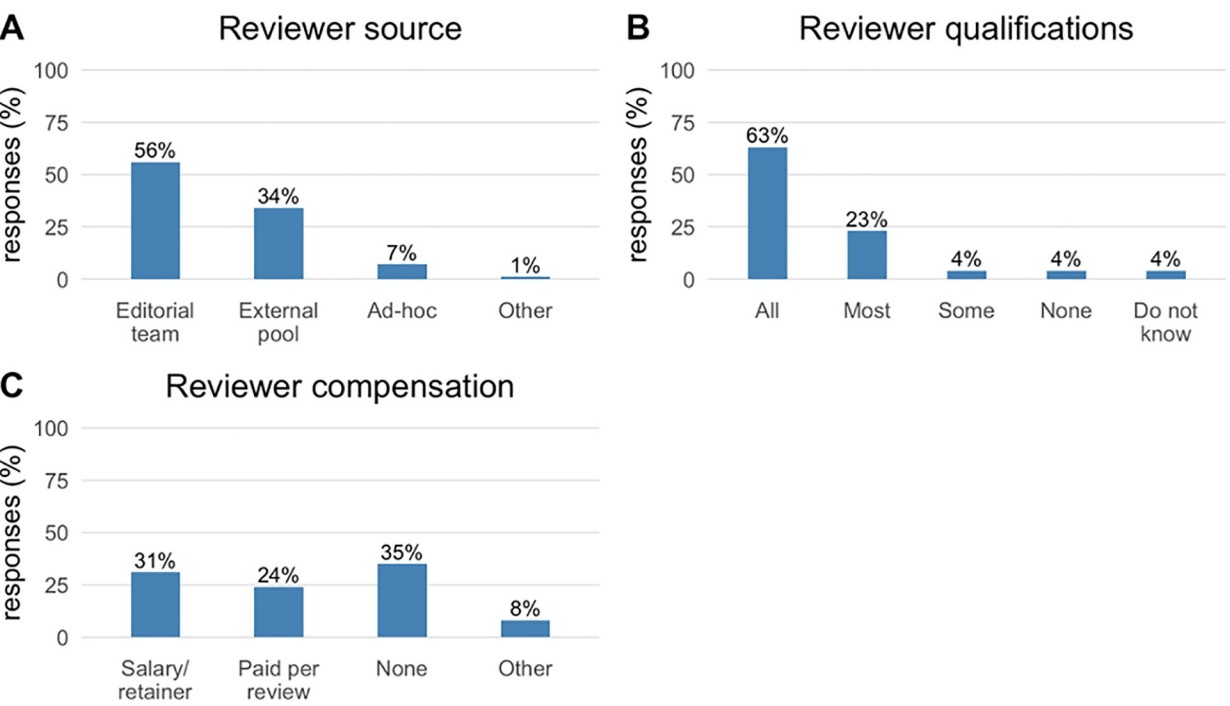

**Fig 2. Statistical reviewer source, qualifications and compensation.** Percentage of responses (N = 71, including 1 missing response for all questions not shown) for questions about policies relating to statistical reviewers. Panel A: 'How are statistical reviewers chosen?' Panel B: 'What proportion have doctoral training level in a quantitative discipline (e.g., biostatistics, epidemiology, informatics, outcomes research)?' Panel C: 'Are statistical reviewers compensated for their work?'.

## Discussion

Concerns about the validity of published scientific claims, coupled with the recognition that suboptimal or frankly erroneous statistical methods or interpretations are pervasive in the published literature, have led to active discussions over the past decade of how to ensure the proper use and interpretation of statistical methods in published biomedical research [2, 7, 25]. Most of the proposed remedies tend to focus on improving study design (e.g., sample sizes), statistical methods, inferential guidance (e.g., the use of p-values), transparency, and statistical training. Comparatively little attention has focused specifically on how journals themselves can improve their performance. The fact that so many problems persist suggests that extant editorial procedures are not adequate to the task [2–8]. That was the motivation for this survey of the highest impact factor biomedical journals across 57 specialties to find out whether and how they use methodologic experts to help them adjudicate and revise manuscripts [9].

The results suggest that although statistical review of some kind is fairly common, it is far from universal; of the 107 eligible journals 34% (36/107) rarely or never use specialized statistical review and 34% (36/107) used it for 10–50% of original research. Only 23% of these top journals subjected all original research to specialized statistical scrutiny. These numbers are quite similar to and no better than those reported in a similar survey by Goodman and colleagues [22] in 1998 where 33% of 114 surveyed biomedical journals employed statistical review for all original research manuscripts and an additional 46% employed statistical review at the editor's discretion. Thus, it seems that there may not have been substantial changes in the use of statistical review over the last 20 years, in spite of the fact that the vast majority of editors in this survey regarded statistical review as having substantial incremental value

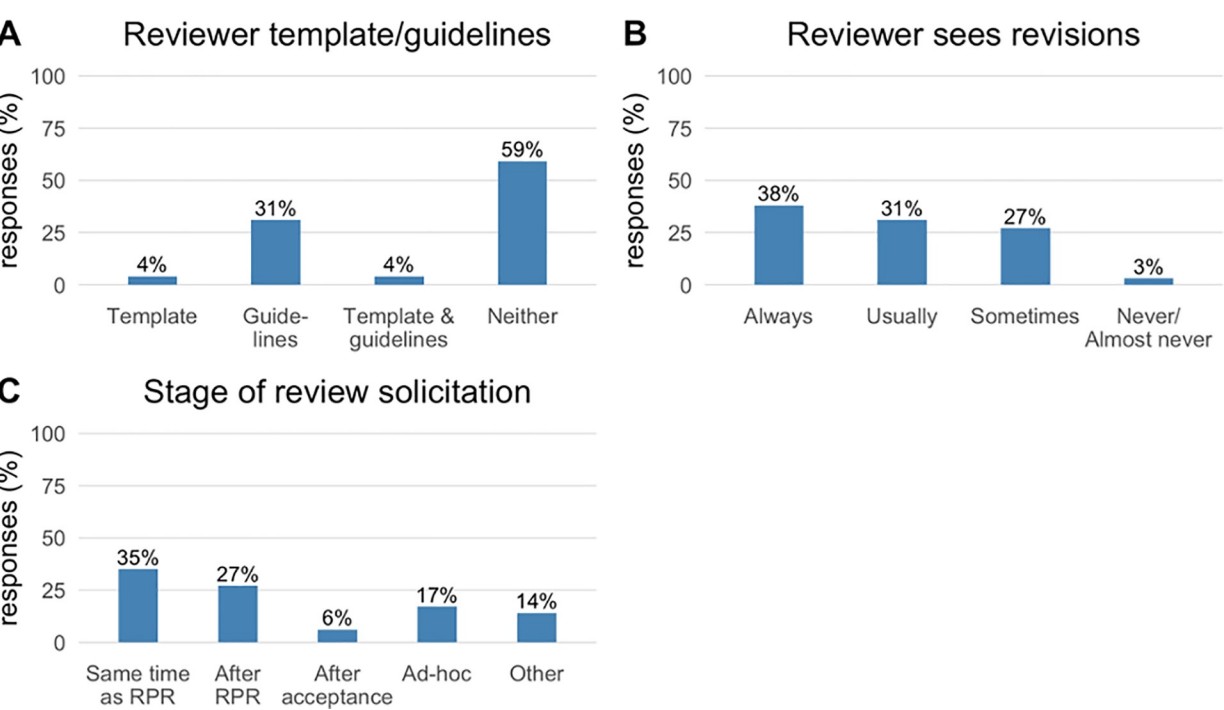

**Fig 3. Journal policies and practices related to statistical reviewers.** Percentage responses (N = 71; including 1 missing response for all questions not shown) for questions about policies relating to statistical review procedures. RPR = Regular Peer Review. Panel A: 'Do you have a formal structure for statistical review that you ask statistical referees to follow?' Panel B: 'How often does the statistical reviewer see a revised version of the manuscript, to assess whether their initial comments were addressed?' Panel C: 'When you do obtain a statistical review, at what stage is the review usually solicited?'.

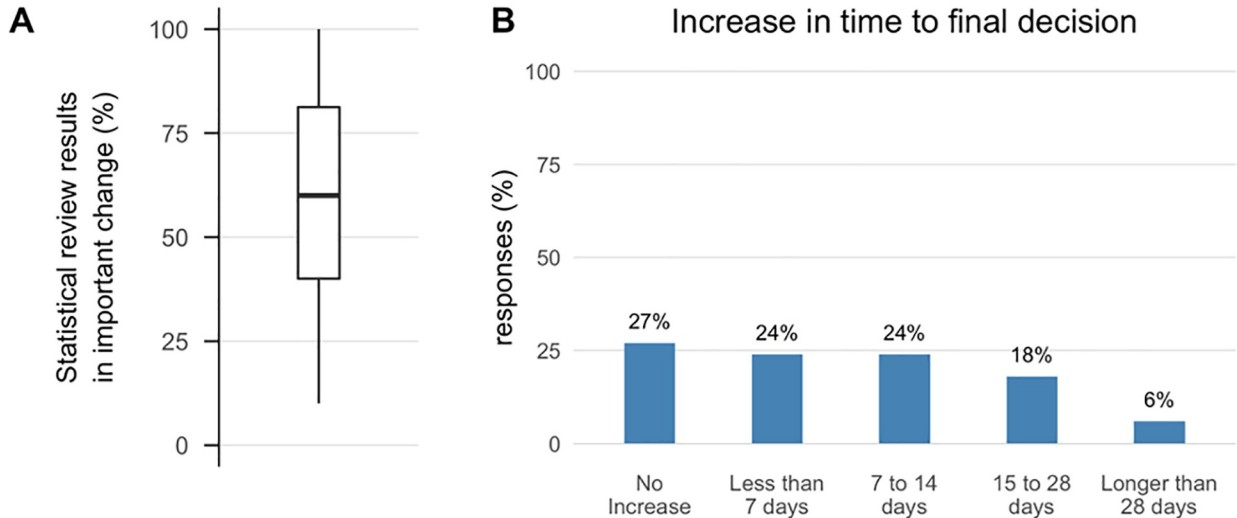

**Fig 4. Statistical review and time to decision.** Percentage of responses (N = 71, including 3 missing responses for Panel A and 1 missing response for Panel B, not shown) for questions about outcomes of statistical policies. For the boxplot the dark horizontal line represents the median, lower and upper hinges correspond to the 25th and 75th percentiles, and the upper and lower whiskers represent the ± 1.5 interquartile range. Panel A: 'When you do obtain a statistical review, approximately what percentage of the time does it result in what you consider to be an important change in the manuscript?' Panel B: 'When you use statistical review, what is the approximate median increase in time to final decision?'.

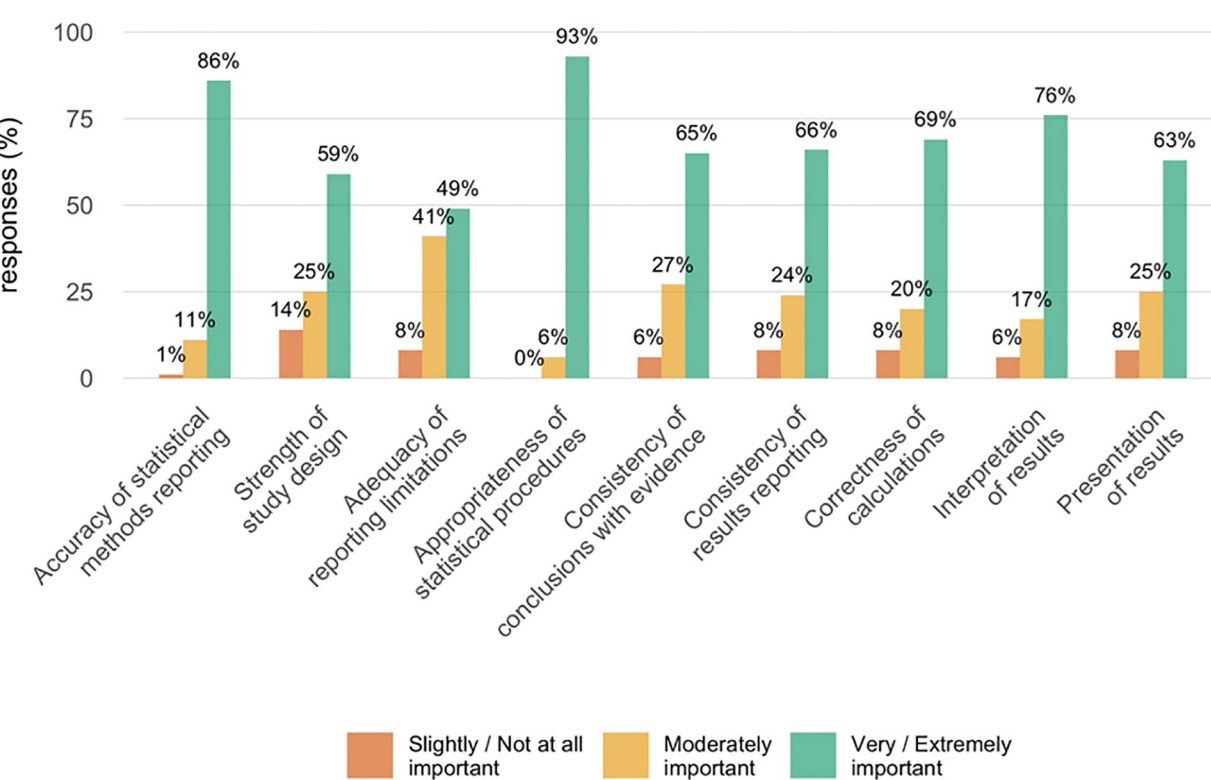

**Fig 5. Incremental importance of statistical review over regular peer review.** Responses to the question 'In your journal, how would you rate the incremental importance of the statistical review (i.e., what it adds to typical peer and editorial review) in assessing these elements of a research report?' N = 71; Percentages do not sum to 100% because of 1–2 missing responses.

beyond regular peer review, improving not just the statistical elements, but interpretation of the results, strength of the conclusions and the reporting of limitations. This impression is supported by empirical assessments of statistical review [10–17]. Interestingly, there was comparatively little concern about the potential increase in reviewing time, cost, and difficulty identifying suitable statistical reviewers.

We did not attempt in this survey to address the quality of statistical review or its implementation, which can vary with the journal model. Adding methodologists to the editorial board may be most effective at facilitating the two-way transfer of knowledge and of journal culture between the statisticians and the other editors [18, 23], improving the methodological sophistication of the entire editorial team over time, and ensuring that reviews target the most critical issues and are communicated and implemented appropriately. This editorial board model was the most common reported here (56%), as it was previously [22]. By contrast, one-third of journals drew their statistical reviewers from an external pool. This model risks using statistical reviewers without adequate domain knowledge, or whose methodologic expertise or preferences are narrow or idiosyncratic. Just like other peer reviewers, individual statistical reviewers have their own limitations, and if there is not a statistician internal to the journal, an editor may not know if statistical reviewer requests are reasonable or how to adjudicate disputes between the statistical reviewer and the authors, who might have a statistician of their own.

Specialized statistical review is just one part of a multistep editorial process. Schriger et al. examined dedicated statistical review at a leading emergency medicine journal, and found that while there was a measurable improvement in statistical quality, a sizable number of errors flagged by statistical reviewers persisted in the published article [17]. This occurred because authors declared they had fixed problems that were not in fact corrected, comments were not transmitted to authors, authors ignored comments, or the author rebutted the comments, all without follow-up by a decision editor. Statistical review processes were subsequently altered to make this less likely at that journal, but this demonstrates that to be effective, the initial statistical review must be enforced by other editorial processes.

Statistical reviewers do not necessarily need to be PhD statisticians; a domain expert with sufficient quantitative training may also take on the role. In our survey, it was reported that most statistical reviewers had doctoral level training in a quantitative discipline, which could include such fields as statistics, epidemiology, informatics, health services research, and economics. About half received financial compensation for their work, somewhat more than the one-third reported Goodman et al. [22]. Unlike other reviewers, financial compensation is often necessary to employ statistical reviewers because they are in wide demand and are not reviewing for their own academic discipline, for which they do not expect compensation. Typically, only the most prominent journals in a field have the resources to pay statistical reviewers, and by targeting high impact-factor journals, this survey may have selected journals most likely to have those resources.

The use of reviewer guidelines or templates was relatively uncommon, as was the case 20 years ago [22]. Guidelines or templates might help to standardize the review process and prompt reviewers to address pertinent statistical issues, improving overall review quality and consistency across reviewers and papers. The Nature journal group has instituted a formal statistical reporting checklist for authors that is electronically linked to the article (https://www.nature.com/documents/nr-reporting-summary.pdf).

This study has several important limitations. Although the absolute number of responses was comparable to those obtained in previous surveys on this topic [21, 22], the response rate (35%) was low enough to be concerned about selection bias, albeit probably towards journals more likely to use statistical review. The survey focused on high impact factor journals, again probably an upwards bias as lower profile journals are unlikely to employ statistical review more frequently [22]. This is supported by a survey of 30 dermatology editors where 24 (80%) rarely used statistical review for original research with data, and only 3 (10%) reported statistically reviewing more than 75% of manuscripts [24]. Only one dermatology journal had an editor primarily responsible for statistics. So, while the fraction of journals (35%) using statistical review for more than half of their articles could be substantially improved, the corresponding number for the non-respondents and for the tens of thousands of other biomedical research journals is probably far lower. Finally, statistical review may be less valuable at review journals of which there were 9 amongst the respondents; we did not explicitly verify whether these journals could potentially benefit from statistical review.

## Recommendations and conclusion

Overall, the findings reported here suggest that statistical review has not dramatically changed at leading biomedical journals over the past 20 years [22] even as concerns about statistical misuse in biomedical research have markedly increased [2]. Most editors seem convinced by the value of statistical review and apply the process to some or sometimes all of the articles that undergo regular peer review.

Efforts to reduce poor statistical practice through statistical review might be best focused on improving standardization, potentially through the provision of guidelines or templates.

Facilitating a more productive two-way dialogue between the statistical and applied research communities may help to mitigate poor practices [7]. Meta-research can be used to elucidate which models of statistical review are more or less effective in different scenarios [25].

New models of peer review and editorial practice might help to address persistent statistical problems in the biomedical literature. Recognizing that statistical review is time intensive, limited by both reviewer supply and expense, perhaps new centralized resources of experienced or vetted methods reviewers could be developed that would supply pre-publication statistical reviews, whose content could be transmitted to any journal to which the paper is submitted. While this might not supplant the statistical reviews at leading journals, it would raise the bar overall for the statistical quality of submitted manuscripts across the publishing landscape. Just as open-access fees of several thousand dollars are now routinely included in federally funded research grants, perhaps a much lower standard fee for independent statistical review could be supported by such funding, which could be used to support the centralized resource, and take the burden off of journals that cannot afford high quality review. Alternatively, either individual journals or their publishers could collectively subscribe to such a service. Review procedures at leading biomedical journals show that even papers with statistician authors can still benefit from independent methodologic review. Finally, it would be critical for such a service to provide feedback to a statistician's home institution, whether it be academic or in the private sector, on the quality and value of their contribution, to provide additional professional incentive to provide such reviews.

The increasing use of open peer review, where all peer review and editorial correspondence is made openly available might help amplify the effect of statistical reviews. Currently, such reviews serve only to improve individual papers, and their content and effect is effectively hidden. Having a public archive of formal statistical reviews could potentially serve as a valuable scientific and didactic resource.

Other models of peer review have been proposed to improve methodologic rigor, but they are unlikely to meet the demand. Pre-print archives and models that promote transparency, code and data sharing and post-publication peer review purport to facilitate the ability of the broader scientific community to probe the cogency of methods and claims. However, while this might indeed be effective for a small proportion of articles, particularly those that garner special attention, it is unlikely to induce change in the vast majority of articles, for which there simply are not enough methodologic readers who will offer in-depth critiques, particularly without incentive to do so. Also, editors use the leverage of possible rejection to require changes that authors might not otherwise accept, but neither preprint nor post-publication review have that leverage. Primary findings and conclusions have much longer lasting effect than ones amended later, as evidenced by retracted articles that continue to be cited, or errata that are ignored, so it is important that the initial publication of record be as accurate as possible.

Given that human expertise is in short supply, what role could artificial intelligence play in improving review of methodologic aspects of a paper? There have been a few attempts to develop programs that examine statistical aspects of a paper, but these are of extremely limited scope, e.g. checking whether the reported degrees of freedom and F or chi-square statistic is consistent with a reported p-value [26], which is mainly of value in the psychological literature, which has a structured way to present such information rarely used in biomedical publications. Some publishers are also experimenting with software that evaluates the use of reporting standards, but other functionality is unclear. [27] Given that methodologic reviewers ideally provide an integrated assessment of the research question, design, conduct, analysis, reporting and conclusions, it is highly unlikely that AI applications will be able to provide substantive help in the near or medium future.

Journal review is only one component of a larger ecosystem that needs changing [25]. Improving the quality of statistical education for researchers and readers of the scientific literature is of paramount importance, particularly in light of documented misunderstanding of foundational statistical concepts in both groups [28]. It is critical to note that statistical education goes well beyond computational training, which is necessary but not remotely sufficient to properly design and analyze research. Research funders are sending this message, with the announcement of new NIH and AHRQ requirements for "rigor and reproducibility" training in T32 grants starting in May 2020 [31]. Training and published research are synergistic; the quality of statistical analyses reported in the highest profile journals creates a de facto standard, sending an important message to young investigators that robust training in statistical reasoning and design will be recognized and rewarded when they submit their research to the best journals in their fields.

## Open practices statement

The study was not pre-registered. All data exclusions, measurements, and analyses conducted during this study are reported in this manuscript. Our survey also included an additional group of psychology journals; however, due differences between the two disciplines, those results are reported elsewhere [29]. All anonymized data (https://doi.org/10.17605/OSF.IO/NSCV3), materials (https://doi.org/10.17605/OSF.IO/P7G8W), and analysis scripts (https://doi.org/10.17605/OSF.IO/DY6KJ) related to this study are publicly available on the Open Science Framework. To facilitate reproducibility, we wrote this manuscript by interleaving regular prose and analysis code, using knitr [30], and have made the manuscript available in a software container (https://doi.org/10.24433/CO.3883021.v2) that re-creates the computational environment in which the original analyses were performed.

## Supporting information

**S1 Table. Number of survey responses by subject area (Web of Science sub-discipline categories for the discipline of biomedicine).**
(DOCX)

**S1 Fig. Histogram showing distribution of estimates for the percentage of original quantitative research articles that undergo statistical review.** The dashed line indicates the $< =$ 10% cut-off point whereby statistical review was considered 'rare' and respondents were redirected towards the end of the survey (see methods section for details).
(DOCX)

## Acknowledgments

We thank Lisa Ann Yu for assistance collecting journal contact details and Daniele Fanelli for discussions about the survey design. We are grateful to all respondents for taking the time to complete the survey.

## Author Contributions

**Conceptualization:** Steven N. Goodman.

**Data curation:** Tom E. Hardwicke, Steven N. Goodman.

**Formal analysis:** Tom E. Hardwicke, Steven N. Goodman.

**Investigation:** Tom E. Hardwicke, Steven N. Goodman.

**Supervision:** Steven N. Goodman.

**Writing – original draft:** Tom E. Hardwicke, Steven N. Goodman.

**Writing – review & editing:** Tom E. Hardwicke.

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
