## [Decision Letter · Decision Letter 0]

28 May 2020

PONE-D-20-12128

How often do leading biomedical journals use statistical experts to evaluate statistical methods? The results of a survey.

PLOS ONE

Dear Dr. Goodman,

Thank you for submitting your manuscript to PLOS ONE. After careful consideration, we feel that it has merit but does not fully meet PLOS ONE’s publication criteria as it currently stands. Therefore, we invite you to submit a revised version of the manuscript that addresses the points raised during the review process.

We look forward to receiving your revised manuscript.

Kind regards,

Despina Koletsi, Dipl.D.S, MSc, Dr. med. dent, MSc, DLSHTM, PGCHE

Academic Editor

PLOS ONE

Journal Requirements:

2. In order to improve reproducibility and replicability, please provide the list of journals surveyed. Please also provide a statement about whether your sample is representative of a larger population.

3. Thank you for stating the following in the Funding Statement Section of your manuscript:

"This work was enabled in part by a fellowship grant to the Meta-Research Innovation Center at Stanford (METRICS) from the Laura and John Arnold Foundation. The Meta-Research Innovation Center Berlin (METRIC-B) is supported by a grant from the Einstein Foundation and Stiftung Charité."

Reviewers' comments:

Reviewer's Responses to Questions

**Comments to the Author**

1. Is the manuscript technically sound, and do the data support the conclusions?

Reviewer #1: Yes

Reviewer #2: Yes

2. Has the statistical analysis been performed appropriately and rigorously? 

Reviewer #1: Yes

Reviewer #2: Yes

3. Have the authors made all data underlying the findings in their manuscript fully available?

Reviewer #1: Yes

Reviewer #2: Yes

4. Is the manuscript presented in an intelligible fashion and written in standard English?

Reviewer #1: Yes

Reviewer #2: Yes

5. Review Comments to the Author

Reviewer #1: The authors conducted a survey to explore how often high impact factor journals in all clinical fields seek statistical expertise for the papers they publish, reasons for not doing so etc. The topic is of general interest and the paper is easy to read.

I am afraid I have not many things to say to improve the paper. There are many issues that are addressed to a good extent.

I expected the situation to be worse but results may undermine the problem. Partly because missing responses are most probably not missing at random (as reported by the authors) and partly because editors believe unconsiously that the problem is dealt with appropriately.

I see a lot of journals asking reviewers whether they have checked the statistical methods or the paper needs someone to check them. No-one checks if the reviewer is knowledgeable of the statistical methods employed in the paper. For example, I am being asked right now by PlosONE "Has the statistical analysis been performed appropriately and rigorously? ".

I liked the comment that external statistician may not be aware of certain statistical methods. I also liked the idea of journals hiring statisticians to work full time for them and it is something I have been wandering for quite some time.

A solution could be to include a statistician in the author list. Cochrane does this for the systematic reviews it publishes. On the other hand, I am aware of many case where a statistician has been asked to be included in the author list just to show that a statistician was used

With the rapid development of easy-to-use software, anyone can do statistical analyses without necessarily understanding its output. The statistical reviewer can probably spot a lot of problems, especially regarding interpretation or methods used but (s)he cannot check whether the method was properly used. As the authors state, the problem is multi-faceted and with the rapidly increasing number of submitted papers and journals emerging, review in general is getting worse.

Reviewer #2: I read with great interest they survey by Harwicke and Goodman entitled "How often do leading biomedical journals use statistical experts to evaluate statistical methods? The results of a survey.". The manuscript is well written and the findings are clearly presented. I have the following 2 comments:

1) I am wondering whether the authors could namely specify the 107 journals in Supplementary Table 1.

2) Considering the journals included in the current and the initial survey 20 years ago, it would be interesting to see graphically the proportion of changes for specific "variables".

6. PLOS authors have the option to publish the peer review history of their article (what does this mean?). If published, this will include your full peer review and any attached files.

Reviewer #1: No

Reviewer #2: No

---

## [Author Response · Author response to Decision Letter 0]

9 Sep 2020

Memo to: PLoS One editors

From: Steven Goodman

Re: Response to reviewers

Manuscript: PONE-D-20-12128

Title: How often do leading biomedical journals use statistical experts to evaluate statistical methods? The results of a survey.

Editors Comments:

1.) “In order to improve reproducibility and replicability, please provide the list of journals surveyed. Please also provide a statement about whether your sample is representative of a larger population.”

We have provided a complete anonymized datafile, with the computational environment, in the OSF framework, as indicated in the Open Science statement of the paper. However, our IRB approved consent form explicitly stated that we would not identify the journals who answered the survey. We described in detail the sampling frame in lines 59-64:

“From the complete list of Web of Science subject categories (228) we identified all 68 sub-domains representing biomedicine. We selected the top 5 journals by impact factor within each sub-domain. We supplemented this list with 68 additional journals previously included in the survey by Goodman and colleagues [22], and assigned each of these to their relevant sub-domain. Finally, we removed any duplicates that appeared in multiple sub-domains. This resulted in a sample of 364 journals.”

S file Senate very jazz but everything very engagedupplementary Table 1, included in the manuscript file shows the number of responding journals in each specialty and sub-specialty. This provides the best evidence we can provide relative to the representativeness of the achieved sample. We do not assert that this is perfectly representative of the whole sample; in fact, as we discuss at length in the manuscript, we believe that the sample is likely to be biased towards more statistical review. This only strengthens the findings, because the use of statistical review in this sample is rather low.

2.) Thank you for stating the following in the Funding Statement Section of your manuscript:

"This work was enabled in part by a fellowship grant to the Meta-Research Innovation Center at Stanford (METRICS) from the Laura and John Arnold Foundation. The Meta-Research Innovation Center Berlin (METRIC-B) is supported by a grant from the Einstein Foundation and Stiftung Charité."

Done.

Reviewer #1: The authors conducted a survey to explore how often high impact factor journals in all clinical fields seek statistical expertise for the papers they publish, reasons for not doing so etc. The topic is of general interest and the paper is easy to read.

I am afraid I have not many things to say to improve the paper. There are many issues that are addressed to a good extent….

The reviewer’s comments are appreciated. S/he provides no comments requiring reply or modification of the paper.

Reviewer #2: I read with great interest they survey by Hardwicke and Goodman entitled "How often do leading biomedical journals use statistical experts to evaluate statistical methods? The results of a survey.". The manuscript is well written and the findings are clearly presented. I have the following 2 comments:

1) I am wondering whether the authors could namely specify the 107 journals in Supplementary Table 1.

For the reasons provided in the response to the editor's comments, we unfortunately cannot do this; the journals were promised confidentiality. mood 

2) Considering the journals included in the current and the initial survey 20 years ago, it would be interesting to see graphically the proportion of changes for specific "variables".

 Unfortunately, we no longer have the data files from the survey 20 years ago. We would have liked to have been able to conduct the same analysis, but we couldn't.

---

## [Editor Report · Decision Letter 1]

10 Sep 2020

How often do leading biomedical journals use statistical experts to evaluate statistical methods? The results of a survey.

PONE-D-20-12128R1

Dear Dr. Goodman,

We’re pleased to inform you that your manuscript has been judged scientifically suitable for publication and will be formally accepted for publication once it meets all outstanding technical requirements.

Kind regards,

Despina Koletsi, Dipl.D.S, MSc, Dr. med. dent, MSc, DLSHTM, PGCHEd

Academic Editor

PLOS ONE
---

## [Editor Report · Acceptance letter]

23 Sep 2020

PONE-D-20-12128R1 

How often do leading biomedical journals use statistical experts to evaluate statistical methods? The results of a survey. 

Dear Dr. Goodman:

I'm pleased to inform you that your manuscript has been deemed suitable for publication in PLOS ONE. Congratulations! Your manuscript is now with our production department. 

Kind regards, 

on behalf of

Dr. Despina Koletsi 

Academic Editor

PLOS ONE